# Domain Adaptation Without the Compute Burden for Efficient Whole Slide Image Analysis

**Umar Marikkar**[1]                                           U.MARIKKAR@SURREY.AC.UK

**Muhammad Awais**[1,2]                                MUHAMMAD.AWAIS@SURREY.AC.UK

**Sara Atito**[1,2]                                              SARA.ATITO@SURREY.AC.UK

[1] *Institute for People-Centered AI, University of Surrey*

[2] *Centre of Vision, Speech and Signal Processing, University of Surrey*

**Editors:** Accepted for publication at MIDL 2026

## Abstract

Computational methods on analyzing Whole Slide Images (WSIs) enable early diagnosis and treatments by supporting pathologists in detection and classification of tumors. However, the extremely high resolution of WSIs makes end-to-end training impractical compared to typical image analysis tasks. To address this, most approaches use pre-trained feature extractors to obtain fixed representations of whole slides, which are then combined with Multiple Instance Learning (MIL) for downstream tasks. These feature extractors are typically pre-trained on natural image datasets such as ImageNet, which fail to capture domain-specific characteristics. Although domain-specific pre-training on histopathology data yields more relevant feature representations, it remains computationally expensive and fail to capture task-specific characteristics within the domain. To address the computational cost and lack of task-specificity in domain-specific pre-training, we propose EfficientWSI (eWSI), a careful integration of Parameter-Efficient-Fine-Tuning (PEFT) and Multiple Instance Learning (MIL) that enables end-to-end training on WSI tasks. We evaluate eWSI on seven WSI-level tasks over Camelyon16, TCGA and BRACS datasets. Our results show that eWSI when applied with ImageNet feature extractors yields strong classification performance, matching or outperforming MILs with in-domain feature extractors, alleviating the need for extensive in-domain pre-training. Furthermore, when eWSI is applied with in-domain feature extractors, it further improves classification performance in most cases, demonstrating its ability to capture task-specific information where beneficial. Our findings suggest that eWSI provides a task-targeted, computationally efficient path for WSI tasks, offering a promising direction for task-specific learning in computational pathology.

**Keywords:** Histopathology, Multiple Instance Learning, Domain adaptation, Parameter-efficient fine-tuning

## 1. Introduction

Computational pathology has gained prominence in assisting pathologists by automating routine observational and analytical tasks in histopathology, thereby facilitating early diagnosis, prognosis assessment, and clinical decision-making. In particular, computational methods using deep learning for Whole Slide Image (WSI) analysis have been developed to automatically detect malignant tumor regions, classify cancer subtypes, and identify molecular properties of tissue, augmenting traditional pathology workflows that rely on manual examination.

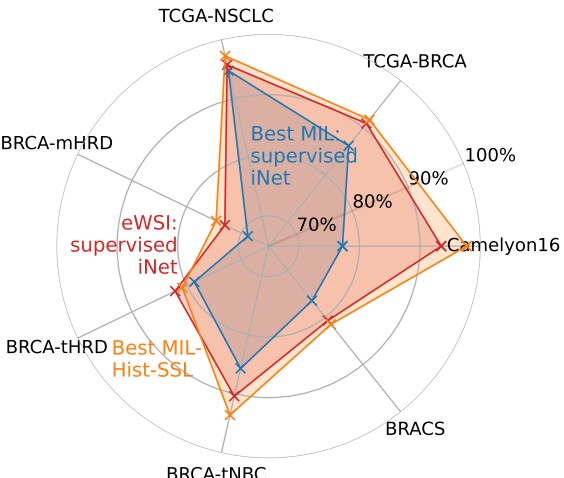

Figure 1: Moving from ImageNet to domain specific encoders, an easy path through eWSI (radial axis is %AUC).

However, traditional deep learning methods for image processing cannot directly be applied on WSIs due to memory constraints. WSIs are extremely large, and high-magnification WSIs can reach sizes on the order of gigapixels (Chen et al., 2022). Therefore, the common approach is to tile WSIs into a large collection of smaller patches (e.g.: ≈ 10k at a tiling resolution of 224 × 224 pixels), pre-compute patch representations through image encoders ahead of time, and perform learning on these fixed patch representations. The latter process of learning from frozen patch representations is typically carried out using Multiple Instance Learning (MIL) methods. In the context of WSI analysis, MIL methods are designed to learn a global slide-level WSI representation by aggregating the collection of individual patch representations using lightweight architectures.

Since MIL methods rely on fixed patch representations, downstream task performance is heavily dependent on the quality of those representations. In spite of that, MIL studies (Ilse et al., 2018; Zhang et al., 2023b; Deng et al., 2024; Shao et al., 2021) typically rely on features obtained from image encoder models pre-trained on ImageNet (Deng et al., 2009), hence their performance on WSI tasks is bottlenecked. While applying MIL methods with in-domain pretrained image encoders (Wang et al., 2022; Kang et al., 2023; Filiot et al., 2023; Lu et al., 2024; Shao et al., 2025) improves downstream performance, such pretraining is computationally expensive. Moreover, even with the use of publicly available in-domain pre-trained encoders, the reliance on fixed patch representations prevents the model from learning task-specific characteristics directly from raw WSI data.

This motivates the exploration of learning protocols that are not bottle-necked by patch representations, while remaining computationally efficient to train. We first build on the observation that existing ImageNet pre-trained encoders already capture low-level features such as intensity levels and texture patterns, which are also relevant in WSI data. Based on this, we investigate an end-to-end framework that reduces reliance on extensive pretraining while allowing task-specific characteristics to be learned directly from raw data. We refer to this framework as **EfficientWSI (eWSI)**. eWSI combines parameter-efficient fine-tuning

(PEFT) with a tailored MIL architecture. In eWSI, the PEFT component adapts encoder weights minimally yet effectively to capture task-specific information at the patch-level. In turn, the MIL architecture is able to leverage the enriched patch-level information to construct a stronger task-specific slide-level representation.

We evaluate eWSI on seven downstream tasks over Camelyon16 (Bejnordi et al., 2017), TCGA (Tomczak et al., 2015), and BRACS (Brancati et al., 2022) datasets. When applied with ImageNet encoders, eWSI outperforms standard MIL-only methods that rely on frozen patch representations extracted from ImageNet encoders, and achieves performance comparable to MIL methods using in-domain patch representations (see Figure 1). Moreover, applying eWSI with in-domain encoders yields further gains on most tasks, indicating that the framework effectively leverages task information from low-level data where beneficial.

## 2. Prior Works

We review existing studies on Multiple Instance Learning, the predominant paradigm for solving WSI tasks; end-to-end learning for WSIs, which is the primary aim of this work; and the emergence of PEFT methods based on low-rank adaptation, which we leverage to enable end-to-end learning.

**Multiple Instance Learning.** A significant body of research exists on MIL methods, particularly for histopathology and standard MIL benchmarks. Ilse et al. (2018) introduce attention-based MIL (ABMIL), using a learned weight vector to aggregate input features. TransMIL (Shao et al., 2021) replaces ABMIL's attention-pooling with a Vision Transformer (ViT) (Dosovitskiy, 2020) variant and Nyström based self attention approximation. Recent advancements include ACMIL (Zhang et al., 2023b), which mitigates overfitting in ABMIL, and AEM (Zhang et al., 2024), which incorporates a loss for attention values itself. IBMIL (Lin et al., 2023) addresses contextual bias using backdoor adjustment, while Snuffy (Jafarinia et al., 2024) combines patch prediction with multi-headed attention. Multi-magnification methods (Li et al., 2021a; Deng et al., 2024; Mirabadi et al., 2024; Thandiackal et al., 2022) show slight improvements but are excluded here. This study focuses on single magnification encoder training, leaving multi-magnification approaches for future work.

**End-to-end processing for WSIs.** End-to-end WSI training research is limited due to the computational constraints caused by WSI size. Early works by Chikontwe et al. (2020) propose Convolutional Neural Networks (CNNs) for joint patch and slide-level classification, selecting top-k patches per epoch for end-to-end training, which is reliant on the pre-computation of all patches at the start of an epoch. Similarly, EPL (Xie et al., 2020) and C2C (Sharma et al., 2021) both leverage patch clustering and use cluster centroid patches to perform end-to-end training. However, they rely on expensive pre-computation—either by clustering all features or per epoch, which incurs significant computational overhead. In contrast, we implement data-efficient patch sampling during training. GIGA-SSL (Lazard et al., 2023) pre-trains WSIs using self-supervised using contrastive learning but freezes slide representations during fine-tuning, limiting task-specific adaptation capabilities. Snuffy (Jafarinia et al., 2024) applies PEFT as pre-training step, but similarly lacks end-to-end learning. Streaming-CLAM (Dooper et al., 2023) combines streaming with CLAM (Lu

et al., 2021) for end-to-end training, achieving high AUC on Camelyon16 but with significant computational overhead (165 seconds per forward pass and 1 hour per epoch on Camelyon16). In contrast, pre-training image encoders on histopathology datasets already achieves comparable performance with less time than StreamingCLAM.

**Parameter Efficient Fine-tuning using low-rank matrices.** DCD (Li et al., 2021b) is one of the initial studies to introduce the idea of low-rank matrices for deep neural networks through dynamic convolutions via matrix decompositions. LoRA (Hu et al., 2021) extend this by freezing pre-trained weights and fine-tuning attention projection matrices in LLMs. Subsequent works (Zhang et al., 2023a; Liu et al., 2024; Koohpayegani et al., 2023) improve efficiency through adaptive rank allocation, separating magnitude and direction changes, or using random basis functions with learnable scalars. In this study, we implement LoRA as the chosen PEFT method, however eWSI can be applied with other PEFT strategies. On integrating PEFT during fine-tuning for WSIs, Lee et al. (2025) perform a thorough benchmark on pathology foundation models, and show that PEFT often outperforms full fine-tuning on WSI downstream tasks. However, their experiments are carried out on large-scale workstations (4× A6000 GPUs) and do not investigate the use of PEFT as a domain adaptation, but rather as a task-adaptation. In contrast, eWSI allows end-to-end training from ImageNet checkpoints on small-scale GPUs, enabled by stochastic sampling, PEFT and the carefully designed MIL aggregator.

## 3. Methods

We first describe the preliminaries, including the WSI processing pipeline and the standard multiple-instance learning (MIL) formulation. We then detail the workings of eWSI.

### 3.1. Preliminaries

**WSI processing pipeline.** We use the terms **WSI** and **patches** to refer to the whole slide image and the local image tiles within the WSI, respectively. For simplicity, we assume that the tiled WSI patches are in a single magnification. Thus, a WSI is represented as,

$$\mathbf{X} = \{x_n \mid n \in N\} \in \mathbb{R}^{N \times 3 \times H \times W} \tag{1}$$

where $x_n$ is a patch at location $n$, $N$ is the number of patches, and $(H, W)$ are the dimensions of the patches within the WSI. Typically, the tiling resolution is set to $224 \times 224$ pixels, consistent with the common practice in computer vision tasks. The goal is to solve the task $P(y \mid \mathbf{X}, \phi)$, where $y = \phi(\mathbf{X})$ is the outcome of the WSI given input $\mathbf{X}$ and model $\phi$. The outcome $y$ is binary in most WSI classification tasks.

We define the overall model $\phi$ as a sequence comprising an image encoder $f(.)$, an aggregator $g(.)$, and a classifier $c(.)$. Given the set of patches $\mathbf{X} = \{x_n \mid n \in N\}$, the image encoder converts the $N$ patches into $N$ local patch representations (2), aggregates the patch representations using $g(\cdot)$, and classifies the aggregated WSI representation using $c(\cdot)$ as,

$$\mathbf{p} = \{f(x_n) \mid n \in N\} \in \mathbb{R}^{N \times D}, \tag{2}$$

$$z = g(\mathbf{p}) \in \mathbb{R}^{1 \times D}, \quad y = c(z) \in \mathbb{R}, \tag{3}$$

where $D$ is the feature dimensionality of $f(\cdot)$, and $y$ is the predicted WSI outcome. The MIL model generally contains both $g(.)$ and $c(.)$.

**The MIL assumption and its relevance to WSI tasks.** The MIL assumption is that, given a 'bag' of 'instances' with unknown instance labels but a known bag-level overall label, a positive bag must contain at least one positive instance, while a negative bag contains only negative instances. Formally, let a bag of instances be denoted as $B = \{x_1, x_2, \ldots, x_n\}$ with label $Y \in \{0, 1\}$, where $Y = 1$ indicates a positive bag and $Y = 0$ a negative bag. The MIL assumption can be expressed as,

$$Y = 1 \text{ if } \exists i \in \{1, \ldots, N\} \text{ s.t. } y_i = 1, \text{ else } 0. \tag{4}$$

where $y_i \in \{0, 1\}$ is the (unknown) label of instance $x_i$. Thus, a positive bag ($Y = 1$) contains at least one positive instance, while a negative bag ($Y = 0$) contains none. In the context of WSI analysis, the MIL assumption translates naturally where each WSI can be regarded as a bag $B$, consisting of a set of patches $\mathbf{X} = \{x_1, x_2, \ldots, x_N\}$ that serve as the instances. The WSI is assigned a slide-level label $Y \in \{0, 1\}$ (e.g., tumor present or absent), while individual patch labels $\{y_i\}$ remain unknown. In this setting, a WSI is labelled positive ($Y = 1$) if at least one patch in the WSI contains tumor tissue ($y_i = 1$ for some $i$), and negative ($Y = 0$) otherwise. Since only bag-level (slide-level) labels are available during training and patch-level annotations are rarely accessible, MIL has emerged as the default paradigm for solving task-specific problems using WSIs.

### 3.2. The eWSI Framework

We implement EfficientWSI (eWSI) to learn task-level WSI representations, without relying on heavy compute resources. eWSI enables on-site training for various downstream tasks on general purpose GPUs. The core components of eWSI are: **(1) Patch sampling:** A strategy for extracting $M$ patches from each WSI at every training iteration, allowing efficient computations and diverse coverage of the WSI's tissue regions across training, **(2) PEFT adaptation:** An encoder with selectively trainable components for effective domain adaptation, and **(3) MIL aggregation:** A gated linear pooling mechanism designed for robust feature aggregation under variable sampling rates, while integrating well with the PEFT encoder. Through this framework, we achieve a balance between efficiency and effective learning, enabling a practical solution for adapting general-purpose models to the task-specific characteristics of WSI data. Below, we elaborate on each component of eWSI.

**Patch sampling and PEFT adaptation.** To reduce computational overhead while keeping the patch encoder $f(\cdot)$ learnable to obtain task-specific patch representations, we perform random patch sampling. To enable effective adaptation of pre-trained encoder weights, we adopt parameter-efficient fine-tuning (PEFT). Specifically, we replace a frozen patch encoder $f(\cdot)$ with a Low-Rank Adaptation (LoRA) encoder (Hu et al., 2021) to fine-tune a frozen backbone on a sparse set of patches sampled from a WSI. The LoRA encoder selectively fine-tunes only the query (`q`) and feed-forward (`mlp`) parameters within the network. This targeted adaptation focuses on updating the components of the encoder that acts as the latent signal (`q`) and the overall knowledge base (`mlp`).

Formally, given the input $\mathbf{X} = \{x_1, x_2, \ldots, x_N\}$, we perform random patch sampling without replacement to obtain a sparse set of patches $\mathbf{X_M}$,

$$\mathbf{X_M} = \texttt{sample}(\mathbf{X}, M) \in \mathbb{R}^{M \times 3 \times h \times w}, \tag{5}$$

where $M$ is the sampled count of patches. For each sampled patch $\mathrm{x}_m$, we pass it through a $\texttt{LoRA}_{\texttt{q,mlp}}$ encoder to obtain the collection of patch representations $\mathbf{p}$.

$$\mathbf{p} = \{\mathrm{p}_m \mid m \in M\} \in \mathbb{R}^{M \times D} \quad \text{where} \quad \mathrm{p}_m = \texttt{LoRA}_{\texttt{q,mlp}}(\mathrm{x}_m). \tag{6}$$

As mentioned above, we adapt only the $\texttt{q}$ and $\texttt{mlp}$ weights of the encoder. We validate the setting of these learnable layers in Section 4.2.

**MIL aggregation.** Given the patch-level representations $\mathbf{p}$, we design a simple yet effective MIL aggregator that integrates well with PEFT and conforms with the core MIL assumption. Traditional attention-based MIL methods, such as ABMIL and ACMIL (Ilse et al., 2018; Zhang et al., 2023b), tend to under perform when the number of sampled patches $M$ is small, which we show later in Section 4.2. In contrast, we introduce a simple MIL aggregator $\texttt{LinMax}$, which is a stack of $L$ fully connected layers with $\tanh(\cdot)$ activations, followed by a max-pooling operation over the patch/instance dimension. This architecture retains the robustness of max-pooling (while also conforming to the MIL assumption) while increasing representational capacity through increased layer depth as opposed to simple max-pooling. Given the encoded patch representations $\mathbf{p} \in \mathbb{R}^{M \times D}$, we compute the global WSI representation $\mathbf{z}$ as,

$$\mathbf{h}^{(l)} = \tanh(\mathbf{h}^{(l-1)}\mathbf{W}^{(l)} + \mathbf{b}^{(l)}) \quad \text{for } l = 1, \ldots, L \quad \text{and} \quad \mathbf{h}^{(0)} = \mathbf{p} \tag{7}$$

$$\mathbf{z} = \max_{m \in \{1, \ldots, M\}} \mathbf{h}_m^{(L)} \in \mathbb{R}^{1 \times d}. \tag{8}$$

Here, $\mathbf{W}^{(1)} \in \mathbb{R}^{D \times d}$ is the projection matrix of the first layer that reduces embedding dimensionality. The subsequent projection matrices $\mathbf{W}^{(2)} \ldots \mathbf{W}^{(L)}$ are isotropic (i.e.: $\mathbb{R}^{d \times d}$). Similarly, dimensionality of the biases follow the same rule with $\mathbf{b}^{(l)} \in \mathbb{R}^D$ and $\mathbf{b}^{(2)} \ldots \mathbf{b}^{(L)} \in \mathbb{R}^d$. After the stacked linear layers, max-pooling is applied element-wise across the $M$ samples. Finally, the obtained global representation $\mathbf{z}$ is passed through a task-specific classifier $c(\cdot)$ to yield the overall WSI outcome. Typically, $c(\cdot)$ is a linear classifier.

The reduced dimension $d$ is chosen such that the overall number of parameters across $\mathbf{W}^{(1)}, \mathbf{b}^{(1)} \ldots \mathbf{W}^{(L)}, \mathbf{b}^{(L)}$ matches a single linear layer with dimension $D$. Formally, given a desired depth of $L$, we solve $Ld^2 + (L + D + 1)d + (-D^2 - D) = 0$ for $d$, where $d$ is then rounded off to the nearest even integer. For example, if $D = 384$ and $L = 3$, this will yield $d = 166$ when rounded off to the nearest even integer. We perform this reduction to explicitly measure the effect of stacking layers without increasing the model capacity via parameter count.

### 3.3. Experimental details

**Datasets, tasks and baselines.** We run experiments on Camelyon16 for tumor metastasis detection, TCGA for cancer subtype classification (IDC vs. ILC in BRCA, LUAD vs. LUSC in NSCLC) and molecular property prediction (mHRD, tHRD, TNBC in BRCA),

and BRACS for 3-way classification (benign, atypical, malignant). We use the published train-test splits for each dataset (Wang et al., 2016; Chen et al., 2022; Lazard et al., 2023; Brancati et al., 2022). For MIL baselines, we reproduce Max-pooling, Max-Pooling followed by an FC layer (FC-Max), ABMIL (Ilse et al., 2018), TransMIL (Shao et al., 2021), and the recent SoTA ACMIL (Zhang et al., 2023b). We also compare against reported results from DSMIL (Li et al., 2021a), IBMIL (Lin et al., 2023), Snuffy (Jafarinia et al., 2024), and AEM (Zhang et al., 2024). We perform experiments using 5 random seeds for Camelyon and BRACS on a single train-test split, and for 5 pre-defined folds for TCGA.

**Model training and evaluation.** We perform our experiments using Vision Transformers (ViT-S/16) (Dosovitskiy, 2020). We use three encoder initializations: **iNet-Sup** which is an image encoder pre-trained on ImageNet using supervised learning, **iNet-SSL** which is an image encoder pre-trained on ImageNet using iBOT SSL (Zhou et al., 2021), and **Hist-SSL**, an in-house image encoder pre-trained on 4 million patches from 20 TCGA cancer datasets using iBOT. We release Hist-SSL in our repository.

We train MIL baselines with 8192 patches per WSI per iteration for Camelyon16, 2048 patches per iteration for BRACS, and 1024 patches per iteration for TCGA. At high sampling rates ($> 1000$), we observe no negative effect on downstream performance, while enabling simpler implementation i.e.: batched inputs. We extract patches at a single objective magnification of 20x, using the OpenSlide library [1]. For eWSI, per single iteration, we sample 64, 384 and 512 patches per WSI on Camelyon16, and 64 patches on TCGA and BRACS. **During inference, we use all available patches**. We perform experiments on RTX 2080Ti (MIL baselines and eWSI$_{64}$) and A100/RTX3090 GPUs (eWSI$_{384,512}$). We further sample 50% of data tokens per patch during the forward pass during training to minimize redundancy. We use a LoRA rank of $r = 4$ and $\alpha = 1$. For `LinMax`, we set $L = 3$.

We use a batch size of 8 for all experiments. We employ AdamW with $\beta = (0.9, 0.999)$ and a weight decay of $5e^{-2}$ for heavy regularization. We tune only the learning rate using k-fold cross-validation on the training sets. We set $lr = 1e^{-3}$ for iNet-SSL and Hist-SSL, and to $lr = 5e^{-4}$ for iNet-Sup, where the lower learning rate ensures stable convergence from a weaker pre-trained checkpoint. A cosine schedule is applied, decaying from $lr$ to $1e^{-6}$, with a linear warm-up from $1e^{-7}$ to $lr$ over the first 5 epochs. We train all models for 100 epochs and perform model ensembling from the final 5 epochs during inference. In practice, convergence typically occurs within 40-50 epochs; however, we observe stable loss throughout training. For consistency, we therefore report all results at 100 epochs, although fewer epochs are sufficient. We release pre-trained models, fine-tuned checkpoints and all code in https://github.com/umarikkar/eWSI.

## 4. Results

### 4.1. Performance of eWSI on downstream tasks

**Tumor metastasis detection on Camelyon16.** We evaluate eWSI and LinMax against state-of-the-art MIL and end-to-end methods on the Camelyon16 test set. As shown in Table 1, LinMax performs similar to other MIL architectures. However, it improves on weaker initializations (i.e.: iNet-Sup) compared to FC-Max, TransMIL and ABMIL. More

---

1. https://pypi.org/project/openslide-python/

Table 1: Performance Comparison (% AUC) on Camelyon16 test split. (*: values reported in literature, ⋆: our standardized implementation. ± 0.0: std not reported.)

| Encoder state | Method | Encoder initialization | | |
| --- | --- | --- | --- | --- |
| | | iNet-sup | iNet-SSL | Hist-SSL |
| Frozen:* | ABMIL | $79.0 \pm 4.9$ | - | $94.5 \pm 2.7$ |
| | CLAM-SB | $76.3 \pm 4.9$ | - | $96.9 \pm 2.4$ |
| | TransMIL | $70.6 \pm 7.6$ | - | $94.3 \pm 0.9$ |
| | DSMIL | $77.3 \pm 3.4$ | - | $91.7 \pm 0.0$ |
| | IBMIL | $79.9 \pm 5.0$ | - | $95.4 \pm 2.2$ |
| | ACMIL | $\mathbf{84.1} \pm 3.0$ | - | $97.4 \pm 1.2$ |
| | Snuffy | - | - | $\mathbf{98.7} \pm 0.0$ |
| | AEM | $80.4 \pm 2.7$ | - | $96.7 \pm 0.8$ |
| Frozen:⋆ | MaxPool | $52.2 \pm 1.8$ | $59.2 \pm 4.7$ | $63.3 \pm 3.2$ |
| | ABMIL | $71.5 \pm 1.3$ | $87.9 \pm 1.6$ | $96.4 \pm 0.8$ |
| | TransMIL | $72.8 \pm 2.9$ | $82.5 \pm 1.9$ | $90.7 \pm 2.9$ |
| | ACMIL | $\mathbf{77.2} \pm 2.1$ | $87.6 \pm 0.8$ | $96.3 \pm 0.4$ |
| | FC-Max | $67.8 \pm 1.3$ | $81.8 \pm 1.8$ | $\mathbf{97.7} \pm 0.6$ |
| | LinMax | $73.9 \pm 2.1$ | $\mathbf{88.0} \pm 1.8$ | $96.7 \pm 0.4$ |
| Adapter / End-to-End | C2C* | - | $91.1 \pm 0.0$ | - |
| | Snuffy-MAE* | - | $91.0 \pm 0.0$ | - |
| | Snuffy-DINO* | - | $93.6 \pm 0.0$ | - |
| | eWSI$_{64}$ | $90.6 \pm 0.8$ | $94.1 \pm 1.1$ | $97.7 \pm 1.2$ |
| | eWSI$_{384}$ | $93.3 \pm 0.6$ | $\mathbf{95.1} \pm 1.7$ | $95.8 \pm 1.2$ |
| | eWSI$_{512}$ | $\mathbf{93.4} \pm 0.9$ | $94.1 \pm 1.4$ | $\mathbf{98.5} \pm 0.7$ |

importantly, LinMax integrates well into PEFT due to its robustness, which we outline in Section 4.2. Moreover, eWSI outperforms all MIL methods, achieving a considerable performance gain with iNet-Sup and 512 patches. This improvement remains significant even with only 64 patches, demonstrating robustness to patch sampling. With an iNet-SSL encoder, eWSI still surpasses the best MIL method. However, using a Hist-SSL encoder (column 3) improves performance slightly, suggesting its frozen features are already well-suited and require little adaptation via PEFT for Camelyon16.

**Cancer subtype classification on TCGA and tumor classification on BRACS.** We evaluate our method on BRACS and TCGA datasets (Table 2), where eWSI consistently outperforms MIL methods across tasks and encoders. We observe improvements using eWSI on the in-domain pre-trained encoder Hist-SSL, indicating that eWSI leverages task-specific learning capacity when available. On BRACS, performance improves with iNet-Sup but drops for iNet-SSL and Hist-SSL, possibly due to a learning plateau, as MIL methods show no gains with Hist-SSL.

### 4.2. Analysis

**The integration of MIL with PEFT.** Table 1 shows that stronger encoders consistently yield better performances regardless of MIL. However, unlike in TCGA, where the tumor region covers approximately 70% of the WSI, the tumor covers approximately 5% of

Table 2: Performance Comparison (% AUC) on TCGA and BRACS datasets.

| | Method | TCGA | | | | | BRACS |
|---|---|---|---|---|---|---|---|
| | | BRCA | NSCLC | mHRD | tHRD | TNBC | |
| Inet-sup | ABMIL | 83.7 ±6.4 | 92.8 ±2.1 | 67.0 ±6.2 | 72.7 ±4.2 | 85.7 ±4.3 | 76.4 ±2.5 |
| | TransMIL | 86.4 ±4.9 | 90.3 ±3.3 | 61.1 ±3.9 | 72.9 ±4.7 | 84.5 ±7.2 | 73.9 ±3.3 |
| | FC-Max | 85.7 ±3.0 | 94.7 ±0.9 | 67.4 ±5.6 | 77.6 ±4.3 | 82.9 ±6.6 | 76.5 ±1.4 |
| | ACMIL | 86.2 ±6.3 | 93.6 ±1.4 | 68.8 ±2.4 | 78.6 ±3.2 | 83.4 ±7.6 | 75.8 ±1.4 |
| | eWSI$_{64}$ | **90.9** ±3.9 | **95.7** ±1.3 | **73.0** ±5.6 | **82.1** ±3.4 | **90.4** ±5.2 | **80.7** ±0.6 |
| Inet-SSL | ABMIL | 87.6 ±3.7 | 92.1 ±1.3 | 67.7 ±3.6 | 71.5 ±4.5 | 82.8 ±4.3 | 77.8 ±4.6 |
| | TransMIL | 90.2 ±5.1 | 93.6 ±1.2 | 63.7 ±5.2 | 75.1 ±4.5 | 76.2 ±10.3 | **82.2** ±1.3 |
| | FC-Max | 91.6 ±3.6 | 95.4 ±1.6 | 69.3 ±2.8 | 78.7 ±4.5 | 89.5 ±3.1 | 80.6 ±2.1 |
| | ACMIL | 89.9 ±4.1 | 94.6 ±0.6 | 68.2 ±4.6 | 71.6 ±3.9 | 86.2 ±9.0 | 81.3 ±0.8 |
| | eWSI$_{64}$ | **93.4** ±4.3 | **96.5** ±1.5 | **72.6** ±1.9 | **81.9** ±4.6 | **92.9** ±4.7 | 81.5 ±1.3 |
| Hist-SSL | ABMIL | 88.9 ±5.0 | 94.7 ±1.5 | 72.0 ±4.1 | 74.9 ±4.9 | 88.9 ±5.7 | 77.9 ±0.5 |
| | TransMIL | 88.3 ±5.9 | 97.1 ±0.8 | 71.2 ±3.3 | 74.9 ±8.7 | 91.3 ±5.4 | 79.2 ±0.3 |
| | FC-Max | 91.7 ±4.1 | 97.2 ±1.3 | 74.6 ±3.6 | 80.8 ±4.8 | 93.6 ±2.9 | 80.7 ±1.7 |
| | ACMIL | 90.1 ±4.0 | 96.5 ±2.1 | 74.1 ±4.3 | 77.5 ±5.6 | 89.7 ±6.3 | **81.5** ±1.9 |
| | eWSI$_{64}$ | **92.4** ±3.8 | **97.8** ±0.8 | **76.3** ±3.4 | **83.7** ±4.9 | **93.7** ±5.3 | 80.7 ±1.0 |

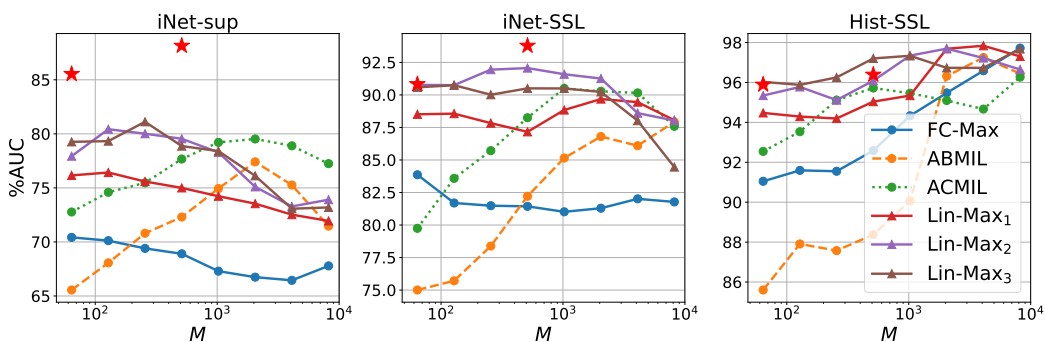

Figure 2: Robustness of MILs to patch sampling for each initialization under frozen encoder settings. ★ denotes the PEFT+Lin-Max$_3$ performance under that setting. Experiments are conducted on Camelyon16, which is particularly sensitive to sampling rate due to smaller regions of interest.

the whole WSI in Camelyon16 (Li et al., 2021a). This poses a difficulty in the attempt to integrate PEFT to the WSI training pipeline, as the compute constraints dictate lesser patches (generally $< 500$ in a practical setting), as opposed to 8192 patches used for MILs with fixed patch representations. This motivates the need to find MIL architectures that are robust to low patch sampling rates to integrate PEFT into training.

We first investigate existing MIL architectures and their robustness to sampling rates, as shown in Figure 2. We find that with fixed patch representation,s attention-based MILs (ABMIL, ACMIL) yield weaker downstream performance with lower sampling rates as opposed to Max-Pooling followed by a linear layer (FC-Max). However, we observe that the overall representation capability of FC-Max is limited as with higher $M$, it fails to outper-

Table 3: Camelyon16 performance vs. $M$ for different MILs with PEFT (seed=0).

| MIL | #M | i-sup | i-SSL | H-SSL |
|---|---|---|---|---|
| ABMIL | 64 | 77.2 | 82.1 | 85.6 |
| | 512 | 72.4 | 90.8 | 94.0 |
| ACMIL | 64 | 86.9 | 83.6 | 83.2 |
| FC-Max | 64 | 85.5 | 90.8 | 95.9 |
| | 512 | 88.2 | 93.8 | 96.4 |
| Lin-Max$_3$ | 64 | 90.6 | 95.8 | **97.7** |
| | 384 | 91.4 | **97.1** | 97.2 |
| | 512 | 93.5 | 96.4 | 97.6 |

Table 4: Camelyon16 performance vs. different trainable layers. Using $M = 64$ and Lin-Max$_3$ (seed=0).

| Layers | % AUC | | |
|---|---|---|---|
| | i-sup | i-SSL | H-SSL |
| $q$ | 84.8 | 92.0 | 96.0 |
| $v$ | 85.4 | 93.4 | 97.0 |
| $q, v$ | 86.7 | 93.6 | **97.8** |
| $mlp$ | 89.5 | 94.3 | 95.9 |
| $q, mlp$ | **90.6** | **95.8** | 97.7 |

form attention-based pooling methods. Assuming layer depth is the limitation, we redesign FC-Max to obtain `LinMax`, by distributing the parameters along stacked linear layers with smaller dimensionality. To increase the feature selecting capability of the network, we introduce tanh activations between the stacked layers. From Figure 2, we find that `LinMax`$_L$ (where **L** is the number of stacked layers) is not only robust but outperforms FC-Max, ABMIL and ACMIL for small $M$, indicating possible integration with PEFT. The trends observed in Figure 2 persist when applying PEFT (see Table 3), where using FC-Max and `LinMax` results in much higher performance overall as opposed to attention-based methods. This highlights the importance of max-pooling to maintain robustness to patch sampling frequency and stacked layers with activations to improve the selective choosing capability. The effect of activation layers is shown in Appendix B.4 in supplementary material.

**Layer and rank assignment for PEFT.** Given that Hist-SSL is pre-trained with iBOT using Inet-SSL initialization, we analyze cross-domain weight transformations, and find that later-layer `mlp` components showed the highest magnitude and direction changes, while `q` and `k` maintained a high degree of change throughout all layers. We provide visualizations for the change of weights in Appendix B.5. This observation in terms of `q` and `k` change is in line with Touvron et al. (2022), where they suggest that fine-tuning `q`, `k`, `v` is sufficient for transfer-learning. However, the study is performed on natural images and does not take into account domain adaptation. We suspect that the change in `mlp` weights can be attributed to the domain shift itself. This led us to adopt `q`, `mlp` as opposed to `q`, `v` in the original study, showing improved performance especially with ImageNet checkpoints (Table 4). However, for Hist-SSL, learning only `mlp` reduced accuracy, suggesting that adapting `q` is enough if the overall knowledge base of the ViT (`mlp`) is sufficient i.e.: domain-specific pre-trained.

We further evaluate the effect of LoRA rank $r$, which governs the size of the reduced space and thereby the overall training capacity of the model. Figure 3 shows that the highest performance is achieved when $r = 4$, and a downward trend is observed with increasing $r$. Further, we find in our experiments that for $r = 384$, which is the equivalent of full fine-tuning for ViTs, the model fails to learn from the weaker iNet-sup checkpoint.

These results indicate that the advantage of LoRA (or PEFT) does not stem from improved training efficiency, as we find that full fine-tuning at $M = 64$ is only slightly slower than PEFT fine-tuning, and still much faster than other end-to-end methods described

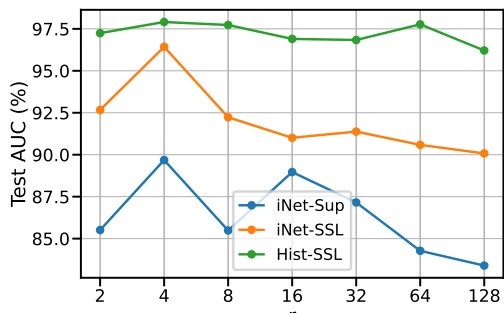

Figure 3: Camelyon16 performance vs. varying LoRA rank $r$. (seed=0).

Table 5: Computational cost for $bs = 1$ on an RTX 3090 GPU. **M**:patch sampling rate. **p**:trainable parameters. $\mathbf{V_{gpu}}$:max VRAM in GB. **t**:time per epoch in mm:ss. $\mathbf{T}$(h): total time for 100 epochs in hours.

| Meth. | $\mathbf{p} \times 10^9$ | **M** | $\mathbf{V_{gpu}}$ | $\mathbf{t}_{mm:ss}$ | $\mathbf{T}$(h) |
|---|---|---|---|---|---|
| Frozen | 0.05 | 8192 | 0.03 | 00:02 | 0.48 |
| eWSI | 0.27 | 64 | 1.10 | 00:15 | 0.42 |
| | | 384 | 6.58 | 01:12 | 2.00 |
| | | 512 | 8.82 | 01:31 | 2.53 |
| C2C | 21.67 | 64 | 1.10 | 25:04 | 40.8 |

later in Table 5. Instead, LoRA provides an implicit regularization through its low-rank constraint, which stabilizes the domain shift and yields a better overall encoder state.

**Computational complexity of eWSI.** We analyze the computational cost of eWSI and compare it against frozen-encoder fine-tuning and runtimes of existing end-to-end WSI methods, in Table 5. Within eWSI, both per-epoch runtime and GPU VRAM scale approximately linearly with the number of sampled patches $M$. As most WSI-level downstream tasks involve relatively small datasets, even a setting of $M = 512$ is practical, as on Camelyon16, training for 100 epochs takes approximately 2.5 hours. A more efficient configuration with $M = 64$ reduces this to 25 minutes. Although the above runtimes exceed those of frozen-encoder fine-tuning, which requires only 3-4 minutes for training, the latter incurs an 26-minute offline feature extraction cost for the training set. When feature extraction is performed online, we observe an epoch time of 8s at $M = 64$, corresponding to ≈13 minutes total training time.

Despite the increased runtime of eWSI, the accuracy gains achieved by eWSI justify this overhead, especially given that the absolute training times remain practical. Compared to existing end-to-end methods, eWSI is substantially more efficient. StreamingCLAM(Dooper et al., 2023) reports 165 seconds per forward pass (batch size 1) on Camelyon16 while performing their experiments on a 40GB V100 GPU. For C2C(Sharma et al., 2021), we calculate the expected training time per epoch with ViT-S, which contains a full feature-extraction step at each epoch for pre-clustering that requires approximately 26 minutes, resulting in an expected total time of 45 hours for 100 epochs.

**Effect of sparse random sampling on downstream inference.** Given that Camelyon16 contains relatively small regions of interest (ROIs), sparse random sampling runs the risk of failing to capture tumor regions during training, producing false positive supervision at the bag level. Interestingly, even in this setting, the primary evaluation metric (AUC) remains high, suggesting that the model is still able to rank test samples according to their likelihood of tumor presence.

However, when applying a fixed probability threshold of 0.5 to compute accuracy, performance drops substantially, as shown in Table 6 for eWSI$_{64}$. For example, when applying eWSI with the iNet-SSL encoder at a sampling rate of $M = 64$, the model achieves an

Table 6: Camelyon16 performance with bias correction. $p_t$: Probability threshold, BC: Manual bias correction, FE: Frozen encoder fine-tuning with dense sampling starting from eWSI encoder features. ACMIL: frozen encoder with $M = 8192$.

| Method | $p_t$ | iNet-Sup | | iNet-SSL | | Hist-SSL | |
|---|---|---|---|---|---|---|---|
| | | AUC | Acc. | AUC | Acc. | AUC | Acc. |
| ACMIL | 0.50 | 78.3 | 72.7 | 90.4 | 79.8 | 96.7 | 90.7 |
| eWSI$_{64}$ | 0.50 | 90.6 | 46.5 | 95.8 | 45.0 | **97.7** | 48.1 |
| | 0.98 | 90.6 | 82.9 | 95.8 | 88.4 | 97.7 | 92.2 |
| eWSI$_{64}$+FE | 0.50 | **91.3** | **83.7** | **96.9** | **94.6** | 96.9 | **93.8** |

AUC of 95.8%, yet the absolute accuracy at a 0.5 threshold is only 45.0%. This can be easily rectified by manually adjusting the decision threshold. Using a threshold of 0.98 increases accuracy to 88.4%, outperforming the accuracy obtained without eWSI under frozen-encoder training with dense sampling (79.8%; see Table 6). This indicates that there is a bias induced by sparse sampling, where predicted probabilities are systematically shifted toward higher values and biased toward the positive class, rather than a failure to discriminate between tumor and non-tumor samples.

This form of manual threshold 'hacking' however requires prior knowledge of dataset-specific statistics. A more elegant solution that avoids manual threshold selection is to perform an additional training run under a frozen-encoder setting using dense sampling with the now-trained eWSI encoder. This incurs a modest computational overhead (the same time required for frozen encoder fine-tuning) and as shown in Table 6, results in higher absolute accuracy without threshold tuning. We provide more insight into the effect of sparse sampling, and observed failure cases in Appendix B.1.

## 5. Conclusion

We propose eWSI, a practical alternative to compute-heavy pre-training to solve Whole Slide Image classification tasks, using a carefully chosen integration of Parameter-Efficient-Fine-Tuning (PEFT) and Multiple Instance Learning (MIL) that supports efficient and effective end-to-end training on WSIs. We first identify the key limitations of existing MIL methods, particularly when sampling a limited number of patches, and address these issues step-by-step, resulting in a robust MIL framework that integrates well with PEFT. For domain adaptation, we study the transformation of pre-trained weights from ImageNet to the Histopathology domain when pre-training, and explore ways to emulate this transfer effectively while preserving source encoder characteristics. We highlight the effectiveness of our approach on Camelyon16, TCGA and BRACS datasets, where we observe significant improvements over ImageNet pre-trained models. Our results demonstrate that eWSI offers a promising and practical alternative for WSI pre-training using a single general purpose GPU, eliminating the need to rely on pre-trained models.

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

## Appendix A. Schematic diagram for eWSI

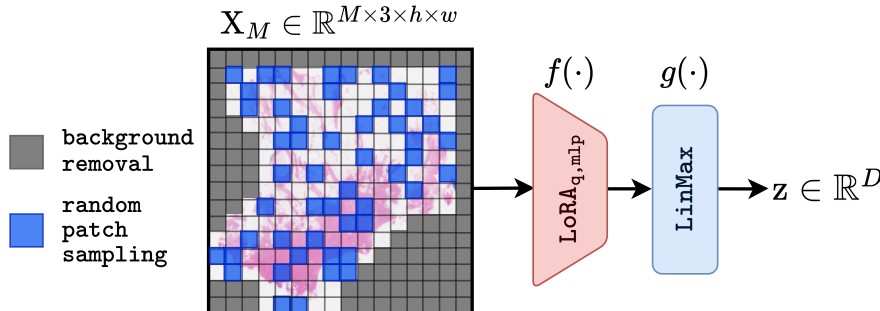

Figure 4: An outline of eWSI. The input patches are randomly sampled and fed into the `LoRA` encoder. The encoded patches are then passed through the LinMax aggregator to yield the global WSI representation $\mathbf{z}$.

## Appendix B. Further analyses

### B.1. Sparse patch sampling and its effect on small regions of interest.

We investigate how sparse sampling during training affects small ROIs in test WSIs. We perform patch-wise classification prior to max-pooling, to obtain patch-wise tumor predictions. The threshold to classify a slide as positive or negative is computed via Youden's thresholding (Youden, 1950). Using the thresholding, we observed 4 false negative slides with eWSI iNet-SSL encoder, shown in Figure 5. We observe that even in the samples wrongly classified as negatives, there exists a high probability of tumor prediction at the true location of the tumor.

We further investigate the relationship between tumor burden and prediction confidence. As the raw prediction probabilities are systematically biased, we focus on the association between the two entities. Specifically, we first compute the tumor percentage as the proportion of annotated tumor area relative to the total tissue area, where tissue is segmented using Otsu's thresholding method (Otsu et al., 1975). We then measure the association between tumor percentage and slide-level prediction probability using Spearman's rank correlation (Schober et al., 2018). We observe a strong positive correlation ($\rho = 0.78$), indicating a significant monotonic relationship between tumor burden and prediction confidence.

### B.2. Sparse patch sampling and its effects on model convergence.

We show the loss curves with sparse and dense sampling in Figure 6. We find that sparse sampling (with or without eWSI) results in a highly volatile training loss, possibly due to the large number of false positives observed during model training. However, this volatility does not translate to volatility in downstream performance, as we observe an acceptable standard deviation under different random seeds, as shown in Table 1.

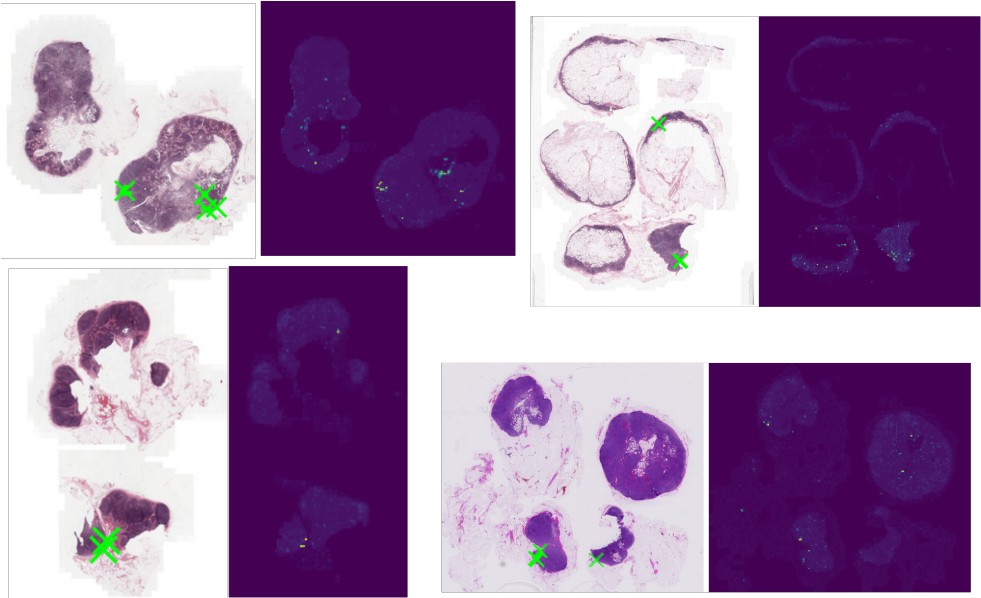

Figure 5: Patch-predictions versus tumor locations for the 4 observed false negatives using eWSI iNet-SSL and $M = 64$. Yellow dots in the heatmaps indicate locations with high probability. The slide IDs are top left:`test_038`, top right:`test_011`, bottom left:`test_013`, and bottom right:`test_099`.

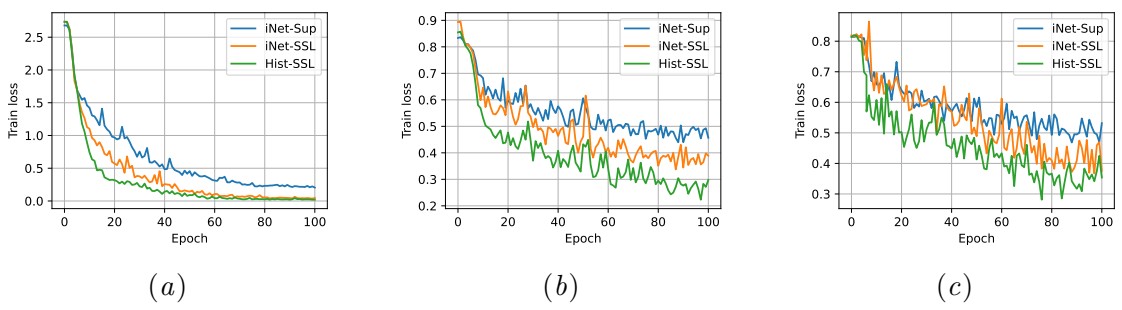

Figure 6: Training convergence under sparse and dense sampling. (a) MIL with $M{=}8192$, (b) MIL with $M{=}64$, and (c) eWSI with $M{=}64$.

### B.3. Experiments with larger encoder backbones.

We run experiments with a larger iBOT ViT-B checkpoint pre-trained on ImageNet (Zhou et al., 2021), and Phikon, a larger histopathology pre-trained checkpoint pre-trained on 40 million TCGA patches (Filiot et al., 2023). Table 7 shows the Camelyon16 results with the aforementioned encoders, comparing between ACMIL under a frozen encoder setting, and eWSI with $M = 64$.

Table 7: Camelyon16 results under different encoder settings. 'Patch' denotes the time taken (under inference mode) to pre-compute patches of the training samples of the downstream dataset.

| Method | AUC % | | Wall time (min) | | |
| --- | --- | --- | --- | --- | --- |
| | iBOT-B | Phikon | Patch | Training | Total |
| ACMIL (Frozen) | $88.08 \pm_{0.7}$ | $99.4 \pm_{0.3}$ | 48.5 | 6.7 | 55.2 |
| eWSI$_{64}$ | $91.61 \pm_{2.2}$ | $99.0 \pm_{0.3}$ | - | 45.8 | 45.8 |

## B.4. The effect of sequential layers and activations for LinMax

When analyzing the effect of sequencing the layers and activations in LinMax, we find that LinMax iteratively attenuates and amplifies certain regions of the WSI. This can be seen in Figure 7.

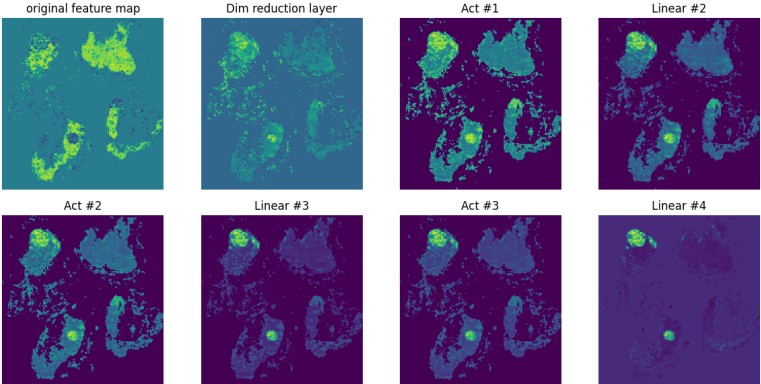

Figure 7: Iterative amplification and attenuation of features in the WSI using LinMax.

To analyze the effect of different activations, we experiment with Tanh, ReLU and Softsign activations, and no activation functions, where we find out that using no activations still achieves better results than FC-Max, perhaps due to the bottle-necking effect caused by lower dimensionality (see Figure 8). Tanh and Softsign being slightly more effective than ReLU indicates the usefulness of a 'gating' mechanism with upper and lower bounds, where the feature values are cut-off at a lower and upper limit as opposed to gating at 0.

## B.5. The domain transfer from ImageNet to Histopathology

As our histopathology pre-trained encoder (Hist-SSL) is pre-trained using iBOT with the initialization being the same iBOT ImageNet encoder (Inet-SSL), we analyze the transformation in weights from the ImageNet to histopathology domain, by plotting the normalized weight magnitude and direction change for each sub-module in each layer, as shown in Figure 9.

We observed that the highest change in magnitude and direction in the latter layers were for the multi-layer perceptron ($mlp$) components in the ViT block. The query and

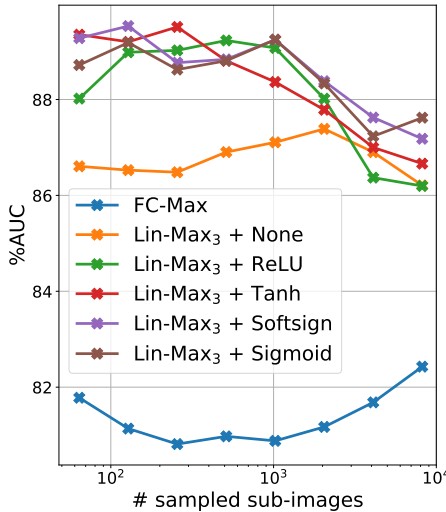

Figure 8: Effect of gate-like activation functions

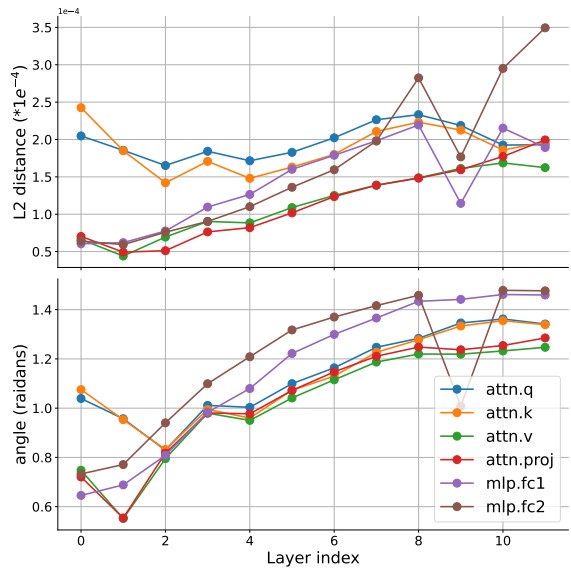

Figure 9: Change of weights in magnitude (top) and direction (bottom) from iNet-SSL to Hist-SSL.

key vectors ($q$ and $k$), maintained a high degree of change throughout all layers. This observation is in line with (Touvron et al., 2022), where the authors suggest that fine-tuning the attention layers is sufficient for transfer-learning. However, the study is performed on natural images and does not take into account domain adaptation. We suspect that the change in $mlp$ weights can be attributed to the domain shift itself.

# Appendix C. Additional visualizations

## C.1. Effect of magnification vs. sampling rate on Camelyon16 performance.



Figure 10: Effect of magnification vs. sampling rate $M$ on Camelyon16 performance. Experiments performing with a frozen encoder setting and ACMIL. horizontal axis is the sampling rate and vertical axis is the magnification

## C.2. Patch representations and pre-pooling activation maps

Figure 11 shows the sub-image representations and the pre-pooling activation maps for iNet-sup, iNet-sup+eWSI and Hist-SSL, respectively. Upon observing the feature maps, we see a higher similarity between the features of iNet-sup+eWSI with the Hist-SSL reference than to the iNet-sup reference, for both encoder features and pre-pooling features. Both the iNet-sup+eWSI and Hist-SSL pre-pooling features show a sparser map in comparison to iNet-sup, and the higher activations tend to be in the samel location.

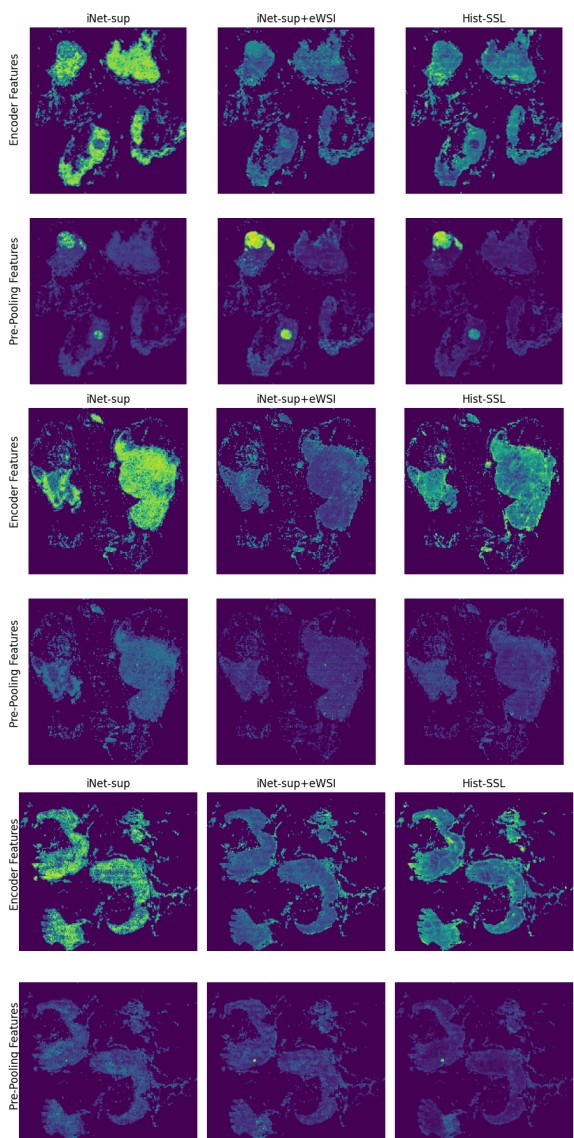

Figure 11: Sub-image representations and the pre-pooling activation maps for iNet-sup, iNet-sup+eWSI and Hist-SSL.

