# OpenReview forum: "Domain Adaptation Without the Compute Burden for Efficient Whole Slide Image Analysis"
_MIDL.io/2026/Conference — MIDL 2026 Poster_

### Official Review · Reviewer_xB8E · 2026-01-04

**Confidence:** 5
**Preliminary Rating:** 3
**Final Rating:** 4

**Summary:**

The paper introduces eWSI, a framework designed to overcome the computational barriers of end-to-end WSI  analysis by integrating PEFT with Multiple Instance Learning. Traditional WSI methods are often bottlenecked by frozen feature extractors pre-trained on ImageNet, while domain specific pre-training remains computationally expensive. To address this, authors utilize LoRA to fine-tune only the query and mlp layers of the vision transformer network using a sparse set of randomly sampled patches. A key contribution is the LinMax aggregator, a series of stacked linear layers with activations & max-pooling, which maintains robustness and high performance even when a small number of patches are sampled (e.g. M = 64).  By evaluating across 7 tasks on Camelyon16, TCGA, and BRACS datasets, the authors demonstrate that their approach when applied with ImageNet feature extractors, can match or outperform MILs with in-domain feature extractors and further improves performance when combined with in-domain feature extractors.

**Strengths:**

- The paper's primary strength is its practicality and accessibility. The framework enables task-specific domain adaptation and end-to-end training on a single general-purpose GPU, such as an RTX 2080Ti, democratizing high-level WSI analysis for researchers with limited compute resources.
- The authors evaluate their method across 7 diverse downstream tasks, including metastasis detection, cancer subtyping, and molecular property prediction, showing consistent performance gains.
- The development of the LinMax aggregator is a clever and scientifically grounded technical contribution. The authors provide convincing evidence that traditional attention-based MIL models struggle when patch sampling is sparse ( e.g. M=64), whereas LinMax, through its stack of linear layers and tanh activations, maintains high performance.
- Additionally, the analysis of cross-domain weight transformations (comparing ImageNet to Histopathology) adds significant value by identifying that adapting the MLP layers is as crucial as the query (q) layers for this specific domain shift.

**Weaknesses:**

- The authors describe Hist-SSL as an encoder pre-trained on 4 million patches from 20 TCGA datasets using iBOT. However, it is not explicitly clarified whether this is an in-house model developed for this study or a publicly available external model. This makes it challenging for others to accurately reproduce the baselines or fully understand the starting state of the domain-specific results.
- The paper does not reference a 2025 study by Lee et al. "Benchmarking pathology foundation models: Adaptation strategies and scenarios." Computers in Biology and Medicine 190 (2025)”. Given that part of this work also explores nearly identical PEFT strategies within the histopathology domain, discussing it would be important for contextualizing the novelty of the eWSI framework.
- While eWSI is frequently described as computationally efficient, this description remains largely qualitative. The paper would benefit from concrete data, such as training time per epoch or peak VRAM usage, to support these claims. Without a normalized comparison of auc versus training time against baselines like Streaming-CLAM or GIGA-SSL on identical hardware, it is difficult to verify the exact efficiency gains achieved.

**Detailed Comments:**

- There is a noticeable lack of detail in the captions for Figure 2, Table 3, and Table 4, which do not specify which dataset the results are referring to. Since the paper evaluates several datasets (Camelyon16, TCGA, and BRACS), specifying the data source within the captions is necessary for clarity & to ensure readers can interpret the results without searching through the main text.
- Several critical details are missing, which significantly impact the reproducibility of the experiments:
1. The study assumes a single magnification, but the specific objective magnification (e.g., 20x or 40x) used for extracting patches from Camelyon16 and TCGA datasets is not specified.
2. While the GPUs used for the experiments are listed, the authors doesn't mention key optimization details like the learning rate, choice of optimizer, or batch size.
3. The authors explain that they adapted the query (q) and mlp parameters using LoRA, but they didn't include the specific rank (r) or alpha (ɑ) values used for the adapter.
4. Similarly, there is mention of the LinMax aggregator using a stack depth of L=3, but the exact hidden dimensions for these fully connected layers aren't specified.

**Justification Of Final Rating:**

I thank the authors for their thorough response and the corresponding updates to the manuscript. My concerns regarding implementation details, clarification of the LinMax aggregation, analysis on stochastic sampling bias have been resolved. Moreover, added computational analysis in Table 5 also seems to demonstrate eWSI’s efficiency, showing that it achieves competitive runtimes compared to frozen feature extraction while enabling end-to-end adaptation. Given these technical clarifications and the improved transparency of the results, I am happy to increase my rating to a Weak Accept

**Justification Of The Preliminary Rating:**

Overall, this paper provides a very much practical answer to one of the important challenges in digital pathology i.e making task-specific model adaptation possible without needing a massive compute, which is achieved here through incorporation of PEFT with MIL. The introduction of the LinMax aggregator is a strong technical addition that prevents the performance drops we usually see when using sparse sampling in MIL. However, the rating is currently as Borderline because there are several highlighted areas where additional clarity & experiments would be beneficial. Addressing those transparency and reproducibility gaps in the final version would significantly strengthen the evaluation.

**Questions To Address In The Rebuttal:**

1. For full reproducibility, would you be able to provide the missing optimization details, such as the batch size, learning rate, etc.
LoRA rank, alpha, and the hidden dimensions used for the LinMax aggregator?
2. To help quantify the efficiency gains of your approach, could you also provide more details on breakdown of training time per epoch and peak VRAM usage for eWSI compared to a standard end-to-end framework like Streaming-CLAM?
3. While the transition to the LinMax aggregator successfully maintains performance on Camelyon16 at a sampling rate of M=64, it would be beneficial to understand the impact of the stochastic sampling process. Could you please provide the standard deviation of the slide-level AUC results when using different random seeds for the M=64 rate? This would provide clearer evidence of the framework's stability and support your claims of robustness.

---

> ### Author Response · Authors · 2026-01-24
> **W1, DC1-4, Q1: Implementation details and reproducability, and W2: Comparison with exisitng PEFT for WSI fine-tuning**
>
> We thank reviewer xB8E for the detailed review of our paper. We also thank the reviewer for acknowledging the primary strength of our work that enables end-to-end training on WSI downstream tasks on general-purpose GPUs. We acknowledge the concerns of reviewer xB8E regarding reproducibility and transparency, and aim to address them below. We denote the reviewers comments as W: weakness, DC: detailed comment, and Q: question.
>
> We group the reviewer's concerns into 4 sections, 1. implementation details, 2. comparison with the existing PEFT method,  3. computational cost, and 4. the effect of stochastic sampling.
>
> ## W1, DC 1-4, Q1: Implementation details and reproducibility
> We aim to address all the reviewer's concerns regarding implementation in this section.
>
> **iBOT pre-trained encoder:**
> - The iBOT encoder is in-house pre-trained. We release the model in our repository along with all details **(patch extraction and training code)** required for reproduction.
>
> **Objective magnification, Hyper-parameter selection and LoRA details:**
>
> - We use an objective magnification of 20x for all our experiments. We mention this in **page 7 of the revised manuscript**, under implementation details. In our preliminary MIL experiments to understand the data, we found that higher magnifications consistently outperform lower magnifications (1,2,5,10), regardless of patch sampling rate $M$ during training. We have added a visualization of how magnification and sampling rate affects downstream performance in **Appendix C.1**.
> - We describe optimizer (AdamW), LR (1e-3 or 5e-4 depending on encoder), batch size (8), WD (5e-2), and scheduling details (cosine with warmup) in **page 7 of the revised manuscript**, under the implementation details section.
> - We use a LoRA rank of $r=4$, and $\alpha=1$, and mention this in **page 7 of the revised manuscript**. We perform further analysis on the LoRA rank $r$, in **page 10 of the revised manuscript**. We find that $r=4$ is optimal, and larger $r$ often results in performance degradation, indicating that LoRA acts as an implicit regularization during training.
>
> **LinMax architecture**:
>
> - We set the hidden depth of LinMax such that, the overall parameter count of LinMax given a certain layer depth, matches the FC-Max parameter count which is a single $D \times D$ linear layer.
> - The architecture of LinMax is such that there is an initial dimensional reduction layer from $D \rightarrow d$, and then a set of isotropic layers of  $d \times d$ separated by $\tanh$ activations, till max-pooling is performed. This reduced dimension $d$ (or the hidden layer dimension) is chosen by solving the following equation:  Given a desired depth of $L$, we solve $Ld^2+(L+D+1)d+(-D^2-D)=0$. For example, if $D=384$ and $L=3$, this will yield $d=166$ when rounded off to the nearest even integer. We perform this reduction to explicitly measure the effect of stacking layers without increasing the model capacity via parameter count.
> - We describe in detail the design of LinMax in **page 6 of the revised manuscript**.
>
> **Writing** : We thank the reviewer for pointing out the missing information in captions. We have fixed the captions in the revised manuscript.
>
> ## W2: Comparison with existing PEFT for WSI fine-tuning:
>
> - As correctly pointed out by the reviewer, [1] performs a thorough benchmark on  pathology foundation models, experimenting with various fine-tuning strategies such as frozen encoder fine-tuning, PEFT and full fine-tuning. However, their experiments are carried out on large-scale workstations (4$\times$ A6000 GPUs) and do not investigate the use of PEFT as an effective method for domain adaptation, but rather as a task-adaptation.
>
> - In contrast, eWSI performs well under end-to-end training on general-purpose GPUs when initialized from ImageNet checkpoints. This is primarily driven by two design choices that do not depend on PEFT: sparse patch sampling and a MIL aggregator that remains stable under sparse sampling regimes. We find that the PEFT component in eWSI mainly acts as an implicit regularizer via its low-rank constraint, rather than a mechanism for reducing computational cost. In practice, the compute reduction from PEFT compared to full fine-tuning is minimal relative to the savings from sparse sampling, since gradients still backpropagate through the full ViT encoder.
> - As this study performs PEFT as a fine-tuning step, we cite this study and discuss the distinction of our work in **page 4 of the revised manuscript**, under the related work section.
>
> 	[1] Jaeung Lee, Jeewoo Lim, Keunho Byeon, and Jin Tae Kwak. Benchmarking pathology foundation models: Adaptation strategies and scenarios. Computers in Biology and Medicine, 190:110031, 2025.

---

> > ### Author Response · Authors · 2026-01-24
> > **W3, Q2: Computational cost, and Q3: Effect of stochastic sampling.**
> >
> > ## W3, Q2: Computational cost
> >
> > We thank the reviewer for this comment. Below, we compare eWSI runtime and GPU memory for Camelyon16 versus frozen encoder fine-tuning and runtimes of existing methods.
> > - In a frozen encoder setting, we define the training time to be the time taken for feature extraction for patches in the training set, and then MIL training. For Camelyon16 feature extraction of the training set, we observed 26 minutes total time for ViT-S. It should be noted that each slide contains ~10k patches with background removal, therefore the time for extraction is costly. However, given the features are extracted, training an MIL on Camelyon16 only requires ~4 minutes total for 100 epochs. Therefore, the total cost can be considered only to be the feature extraction cost.
> > - On the other hand, eWSI with $M=64$ requires approximately 15 seconds per training epoch, without any feature extraction cost. This equates to 25 minutes total training time, as only ($64 \times \text{batch size}$) patches are extracted on-the-fly during a single training iteration.
> >
> > -	When comparing with existing end-to-end methods, StreamingCLAM reports a runtime of 165 seconds per forward pass, making a direct comparison unnecessary. Therefore, we do not include StreamingCLAM in our efficiency comparisons. The closest method in terms of efficient end-to-end training is C2C, which only passes a fixed number of pre-defined patches during training (not random). However, C2C still requires extracting features from the full training set at the start of each epoch, which introduces the additional overhead mentioned above.
> > -	Inference cost remains the same across methods, since all approaches ultimately need to extract features from all patches of the test WSIs for evaluation, regardless of whether this extraction is performed before or after training.
> >
> > - We add a section on the computational cost of eWSI in **page 11 of the revised manuscript**. We also tabulate the results in **Table 5 of the revised manuscript**, and its copy is shown below.
> >
> > 	 Compute cost comparison.
> > 	 p × 10⁹: number of parameters; M: number of patches; V_gpu: peak GPU memory (GB); t_ep: time per epoch in mm:ss; T: total time for 100 epochs in hours.
> > 	 | Method   | p × 10⁹ | M    | V_gpu | t_ep   | T(h)
> > 	 |----------|---------|------|-------|--------|--------|
> > 	 | Frozen   | 0.05    | 8192 | 0.03  | 00:02  | 0.48  |
> > 	 | eWSI    | 0.27    | 64   | 1.10  | 00:15  | 0.42  |
> > 	 | eWSI    | 0.27    | 384  | 6.58  | 01:12  | 2.00  |
> > 	 | eWSI    | 0.27    | 512  | 8.82  | 01:31  | 2.53  |
> > 	 | C2C     | 21.67   | 64   | 1.10  | 25:04  | 40.8  |
> >
> >
> > ## Q3: Effect of stochastic sampling:
> >
> > **The standard deviation values for eWSI M=64:**:
> >
> > - We provide the standard deviation results for eWSI $M=64$, in **Table 1 in the original and revised manuscript**. All values in table 1 are computed for 5 random seeds. We observe standard deviations of 0.8%, 1.1% and 1.2% for INet-sup, iNet-SSL, and Hist-SSL encoders, respectively.
> > -   We would like to clarify an implementation detail about 5-seed evaluation and k-fold cross-validation. We perform k-fold cross validation by splitting the training set, to choose the optimal learning rate. We then perform training for 5 random seeds  on the original train-test split. We do not perform any early stopping, and simply let the model train for 100 epochs for consistency. We have clarified this in **page 7 of the revised manuscript**.
> >
> > **Further impacts of stochastic sampling:**
> >
> > - We further describe in detail, the impact of stochastic sampling in **page 10,11 of the revised manuscript**.
> > - To summarize, we find a systematic bias learnt during training with sparse random sampling. We outline methods to mitigate this bias, resulting in both increased AUC which is independent of prediction bias, and absolute accuracy which is dependent on prediction bias.

---

### Official Review · Reviewer_pkPy · 2026-01-09

**Confidence:** 3
**Preliminary Rating:** 3

**Summary:**

The paper proposes Efficient WSI (eWSI), a framework designed to enable end-to-end training of Whole Slide Image (WSI) models without the prohibitive computational costs usually associated with such tasks. The authors integrate Parameter-Efficient Fine-Tuning (PEFT), specifically Low-Rank Adaptation (LoRA), into the MIL pipeline, allowing a standard ImageNet pre-trained Vision Transformer (ViT) to be fine-tuned on a sparse, random subset of patches ($M=64$ to $512$) per iteration. To address the performance degradation of traditional attention-based MIL aggregators under such sparse sampling, the paper introduces LinMax, a robust aggregator consisting of stacked linear layers with tanh activations followed by max-pooling. Experiments on Camelyon16, TCGA, and BRACS demonstrate that eWSI with standard ImageNet encoders can match or outperform state-of-the-art MIL methods that rely on computationally expensive, domain-specific pre-trained encoders (like Hist-SSL).

**Strengths:**

1. The application of LoRA to bridge the domain gap between natural images (ImageNet) and histopathology is well-motivated and executed. The results convincingly show that adapting a general encoder via PEFT is a highly efficient alternative to full domain-specific pre-training.
2. The paper identifies a critical failure mode of standard Attention-MIL methods: their inability to perform well with sparse patch sampling. This observation provides a strong, data-driven justification for the design of the proposed LinMax aggregator.
3. The method demonstrates consistent performance gains. The proposed eWSI with a generic ImageNet initialization competes with or beats baselines using encoders trained on millions of histology images, highlighting the efficacy of the proposed fine-tuning strategy.

**Weaknesses:**

1. The reliance on sampling a very small number of patches (e.g., $M=64$) is risky for "needle-in-a-haystack" tasks like metastasis detection (Camelyon16). If a positive slide has a tiny tumor burden (<1%), random sampling will frequently yield a bag of negative patches labeled as "positive" during training. This introduces significant label noise. The paper lacks a detailed analysis of how this impacts training stability or if it necessitates a significantly higher number of epochs to converge compared to standard MIL approaches.
2. While "LinMax" is effective, it is architecturally very similar to the standard Deep MIL with Max-Pooling approaches used in earlier computational pathology works (e.g., Campanella et al., Nature Medicine 2019). The novelty lies more in its specific application to the sparse sampling regime rather than the architecture itself.
3. The paper does not quantify the operational training time of eWSI compared to the standard "Extract Features $\rightarrow$ Train MIL" pipeline. End-to-end training, even with LoRA, involves backpropagation through the ViT, which is computationally heavier per iteration than training a lightweight MIL on pre-extracted features.

**Detailed Comments:**

1. In Equation 1, $N$ is used for the number of patches, but Equation 5 uses $n$ for the number of instances.
2. Please specify the rank $r$ used for the LoRA adapters in the implementation details section.
3. Does the inference stage also use random sampling, or does it process the whole slide? If it processes the whole slide, how does the domain shift between "sparse training statistics" and "dense inference statistics" affect the Batch Normalization layers (if any) or the aggregator?

**Justification Of The Preliminary Rating:**

The paper presents an interesting observation and the proposed method shows good performance on several benchmark. However, the paper needs several clarifications of details, in particular its novelty compared to previous works.

**Questions To Address In The Rebuttal:**

1. For Camelyon16, given the small tumor sizes, the model likely receives "clean" inputs (correctly labeled bags) only rarely when $M=64$. Did you observe instability in the loss function? How does the convergence rate (number of epochs) compare to standard MIL?
2. How does LinMax differ from the max-pooling based MIL baselines established in prior literature (e.g., Campanella et al. 2019)? Is the "stacked" aspect (multiple linear layers) the key differentiator?

---

> ### Author Response · Authors · 2026-01-23
> **W1, DC3, Q1: concerns on sparse sampling.**
>
> We thank reviewer pkPy for the detailed review, and acknowledging the strengths of our paper. We acknowledge reviewer pkPy's concerns on sparse sampling, compute cost, and MIL architecture, and to address each of the concerns below. We denote each comment as W: weakness, DC: detailed comment, and Q: question.
>
> ## W1, DC3, Q1: Sparse sampling.
> We thank the reviewer for the comments on sparse sampling, and giving us the opportunity to investigate this further. We address the reviewer's concerns as follows.
>
> **Sparse training vs. dense inference:**
>
> - During training, we perform random patch sampling, and during inference, we forward pass all the patches. We apologize for not making this clear, and we have fixed this in **page 7 of the revised manuscript**, under the implementation details section. We have also added the necessary implementation details needed for reproduction, along with the link to pre-trained models and code.
> - On the statistics affecting aggregator layers, the aggregator layers do not contain any normalization components. They are a sequential set of mix$\rightarrow$gate$\rightarrow$mix$\rightarrow$gate ...  layers, where the features for the next step are mixed through a linear layer, and then gated through a tanh (or any gating) layer. The domain shift between "sparse training statistics" and "dense inference statistics" induces a systematic bias in predicted outcomes, and can be easily rectified. We discuss this systematic bias and steps taken to solve this bias below, and in **pages 11, 12 in the revised manuscript**.
>
> **On sparse sampling creating false positives:**
>  - We agree that a number of false positives occur during training under sparse sampling. However, the common metric used to measure performance, which is the AUC, considers relative ranking of instances. We find that eWSI with sparse sampling during training is still able to build a hierarchy of which slides are more likely to contain tumors, with > 90% AUC across all three encoders. However, as expected, the prediction accuracy is very low at ~50%, when a threshold of 0.5 is used for prediction. This can be easily rectified by changing the threshold to a large value (0.98, resulting in > 90% ACC), indicating that during training, only a bias is induced to the prediction probabilities which are systematically shifted towards 1, rather than a failure to discriminate between tumor and non-tumor.
> - However, this requires manual threshold setting. A more elegant solution is to re-train under a frozen-encoder setting with dense sampling (8192 patches), but using the new eWSI sparse-sample trained encoder as the feature extractor. This simple step (which is cost-effective, as it is frozen-encoder trained) mitigates the inherent bias caused by sparse sampling during training. We show results with manual threshold assignment, and light-weight frozen-encoder re-training in **Table 6** in the revised manuscript (as shown below), along with the explanation on Effect of sparse random sampling on downstream inference, in **pages 11 and 12 of the revised manuscript**.
>
> 	**Table 6:** Camelyon16 performance with bias correction.
> 	 *pₜ*: probability threshold. **FE**: frozen-encoder fine-tuning with dense sampling starting from eWSI encoder features.
> 	 ACMIL corresponds to frozen-encoder performance with *M = 8192*.
>
> 	 | Method      | pₜ  | iNet-Sup AUC | iNet-Sup Acc. | iNet-SSL AUC | iNet-SSL Acc. | Hist-SSL AUC | Hist-SSL Acc. |
> 	 |-------------|-----|--------------|---------------|--------------|---------------|--------------|---------------|
> 	 | ACMIL       | 0.50| 78.3         | 72.7          | 90.4         | 79.8          | 96.7         | 90.7          |
> 	 | eWSI₆₄      | 0.50| 90.6         | 46.5          | 95.8         | 45.0          | **97.7**     | 48.1          |
> 	 | eWSI₆₄      | 0.98| 90.6         | 82.9          | 95.8         | 88.4          | 97.7         | 92.2          |
> 	 | eWSI₆₄ + FE | 0.50| **91.3**     | **83.7**      | **96.9**     | **94.6**      | 96.9         | **93.8**      |
>
> **Training stability and convergence:**
> - We note that training at $M=64$ contains some volatility, regardless of LoRA adaptation or frozen encoder fine-tuning. With full fine-tuning, convergence is highly sensitive to hyperparameters, and even when converged, we find that performance is lower.
> - Further, with both LoRA and frozen-encoder MIL, convergence depends on both the pre-trained encoder state and the sampling rate. We show the loss curves vs. sampling rate and encoder state in **Appendix B.2 in the revised manuscript**.

---

> > ### Author Response · Authors · 2026-01-23
> > **W2, Q2: LinMax aggregator and the effect of stacking layers.**
> >
> > ##  W2, Q2: LinMax aggregator and the effect of stacking layers.
> > We thank the reviewer for this comment. The short anwer to Q2 is partly yes, but it is the joint combination of stacking and max-pooling that contribute to increased performance. As correctly pointed out, the main advantage of LinMax is its effectiveness during sparse sampling, thereby enabling end-to-end training. We address the reviewer's concerns in detail below, first starting with the prior work that performs max-pooling [1], and then on the effectiveness of stacking layers.
> >
> > **Comparison versus exsiting methods:**
> >
> > - The method introduced by [1] is a two-stage classification strategy. At the first stage, they first train a patch-wise linear classifier, where each patch is first reduced from its original representation to a single scalar. Then max-pooling is applied where the highest scalar patch is used for the loss. For the second stage, they re-use these scalar values to obtain top S patches, and the representations of these top S patches are passed through an RNN-based network which is trained to obtain the final score.
> > - However, individual patch prediction (the first stage) in itself, performs worse on Camelyon16 where tumor regions are small, as noted by [2], with 0.54 AUC. In contrast, later pooling approaches rely on a **process**$\rightarrow$**pool**$\rightarrow$**process** paradigm. For example, ABMIL and ACMIL first perform weighted pooling, and then process the features for classification.
> >
> > **Main contribution of LinMax:**
> > - Our contribution lies in how to *best prepare features for pooling*, given a sparse sampling regime. Our initial implementation of this is FC-Max, which performs a single linear projection, then max-pooling, and then classification. Our ABMIL and ACMIL implementations also contain linear projections before pooling (as implemented in their original work), but FC-Max outperforms these methods with $M=64$, suggesting max-pooling is more robust to sparse sampling as opposed to weighted attention pooling.
> > - We then propose LinMax, which **stack** the layers with **gated    activations**, and show that LinMax performs exceptionally well under  sparse sampling regimes, as shown in Figure 2 in the original and revised manuscript. An alternative to LinMax would be stacking the layers with activations as above, and performing weighted-pooling    (i.e.: ACMIL) instead of max-pooling. We perform this experiment with eWSI on Camelyon16, $M=64$ patches for seed=0, and observe AUC=75.5% for iNet-Sup, AUC=78.2% for iNet-SSL, and AUC=97.6% for Hist-SSL, which are worse than max-pooling. This indicates that max-pooling is important to maintain robustness to sparse sampling, while stacked layers contribute in preparing features prior to pooling.
> >
> > [1] Campanella, Gabriele, et al. "Clinical-grade computational pathology using weakly supervised deep learning on whole slide images." _Nature medicine_ 25.8 (2019): 1301-1309.
> >
> > [2] Li, Bin, Yin Li, and Kevin W. Eliceiri. "Dual-stream multiple instance learning network for whole slide image classification with self-supervised contrastive learning." Proceedings of the IEEE/CVF conference on computer vision and pattern recognition. 2021.

---

> > ### Author Response · Authors · 2026-01-23
> > **W3: Compute cost comparison, and DC1, DC2: minor corrections.**
> >
> > ## W3: Computational cost.
> >
> > We thank the reviewer for this comment. We agree that it is computationally heavier to back-propagate through the encoder, as opposed to training an MIL on fixed features. Notably, using PEFT only slightly reduces the computational overhead over full fine-tuning, and the main contributor for fast training is the random sampling. However, we note that the increased cost is still practical as opposed to existing end-to-end methods, which we show below.
> >
> > - We compare eWSI runtime and GPU memory for Camelyon16 versus frozen encoder fine-tuning and runtimes of existing methods. For Camelyon16 feature extraction of the training set, we observed 26 minutes total time for ViT-S. It should be noted that each slide contains ~10k patches with background removal, therefore the time for extraction is costly. However, given the features are extracted, training an MIL on Camelyon16 only requires ~4 minutes total for 100 epochs. Therefore, the total cost can be considered only to be the feature extraction cost. On the other hand, eWSI with $M=64$ requires approximately 15 seconds per training epoch, without any feature extraction cost. This equates to 25 minutes total training time.
> >
> > -	When comparing with existing end-to-end methods, StreamingCLAM reports a runtime of 165 seconds per forward pass, making a direct comparison unnecessary. Other approaches, such as C2C, use sparse sampling but still require extracting features from the full training set at the start of each epoch, which introduces the additional overhead mentioned above. Inference cost remains the same across methods, since all approaches ultimately need to extract features from all patches of the test WSIs for evaluation, regardless of whether this extraction is performed before or after training.
> >
> > - We add a section on the computational cost of eWSI in **page 11 of the revised manuscript**. We also tabulate the results in **Table 5 of the revised manuscript**, and its copy is shown below.
> >
> > 	 Compute cost comparison.
> > 	 p × 10⁹: number of parameters; M: number of patches; V_gpu: peak GPU memory (GB); t_ep: time per epoch in mm:ss; T: total time for 100 epochs in hours.
> > 	 | Method   | p × 10⁹ | M    | V_gpu | t_ep   | T(h)
> > 	 |----------|---------|------|-------|--------|--------|
> > 	 | Frozen   | 0.05    | 8192 | 0.03  | 00:02  | 0.48  |
> > 	 | eWSI    | 0.27    | 64   | 1.10  | 00:15  | 0.42  |
> > 	 | eWSI    | 0.27    | 384  | 6.58  | 01:12  | 2.00  |
> > 	 | eWSI    | 0.27    | 512  | 8.82  | 01:31  | 2.53  |
> > 	 | C2C     | 21.67   | 64   | 1.10  | 25:04  | 40.8  |
> >
> >
> >
> > ## DC1, DC2: Minor corrections.
> > - We have fixed the nomenclature in Eq. 5 (now Eq. 4 as the MIL assumption equation has been shortened).
> > - We have specifcied the LoRA rank $r=4$ in the implementation details, and provided further
> >    analysis on the effect of $r$ on downstream performance in **page 11 of the revised manuscript**.

---

### Official Review · Reviewer_P5C2 · 2026-01-09

**Confidence:** 4
**Preliminary Rating:** 3
**Final Rating:** 4

**Summary:**

The paper proposes eWSI, the first method that couples LoRA with Multiple-Instance Learning (MIL) to enable end-to-end training on Whole-Slide Images (WSIs) with only a single consumer GPU. Instead of freezing ImageNet features (domain gap) or performing expensive histopathology pre-training, eWSI randomly samples 64–512 patches per slide, fine-tunes only the q-projection and MLP weights of a ViT-S/16 with LoRA, and aggregates patch tokens by a lightweight “LinMax” block (3×FC+tanh+max-pool). Evaluated on Camelyon16, TCGA and BRACS, eWSI consistently outperforms several MIL baselines and reaches or exceeds the performance ceiling obtained by large-scale histopathology self-supervised pre-training while using <4 % trainable parameters and no extra unlabeled histopathology data.

**Strengths:**

1. integrate parameter-efficient fine-tuning into WSI end-to-end training.
2. Strong empirical gains: +8–15 AUC on ImageNet init, matches or beats Hist-SSL with 50× less compute.
3. Thorough ablation on sampling robustness and which layers to adapt.

**Weaknesses:**

1. Only ViT-S/16 tested; unclear if findings transfer to CNN backbones or larger ViT.
2. LoRA rank r and number of sampled patches M are not jointly ablated; efficiency frontier unknown.
3. LinMax design justification incomplete: no comparison with “max+light-attention” hybrid.

**Detailed Comments:**

1. Please explain why in Table 1, some of the std value is 0.0.
2. The abstract promises “without the compute burden”, yet main paper gives no training / inference time, GPU peak memory, or energy. It would be better to report total hours to convergence on Camelyon16;  comparison with frozen-feature MIL (same GPU).
3. Does LoRA+q,mlp still win with ResNet50 or the recent pathology-specific Uni/Phikon encoders?
4. Camelyon16 tumour area can be <1 %. Show 2–3 failure slide IDs, their tumour percentage, and attention heat-maps. Quantify localisation FROC if possible; this clarifies whether errors come from domain shift or insufficient sampling.

**Justification Of Final Rating:**

The authors have addressed my primary concerns regarding computational efficiency and model generalizability. I am satisfied with the added analysis on failure modes and the justification for the LinMax aggregator. Overall, this is a solid contribution that provides a practical, low-cost solution for end-to-end WSI training.

**Justification Of The Preliminary Rating:**

The paper opens a promising research direction (PEFT for gigapixel images) and shows impressive empirical gains with minimal resources.  It would be better if the paper provides more statistical rigor, efficiency metrics, and ablation depth.

**Questions To Address In The Rebuttal:**

1. Report total training time, GPU peak memory and energy for eWSI64 vs frozen ACMIL on the same hardware.
2. Detail failure modes: relationship between tumour percentage and prediction confidence.

---

> ### Author Response · Authors · 2026-01-24
> **W1, DC3: Findings under larger ViT and DC1: std values being 0.0**
>
> We thank reviewer P5C2 for their review, and for acknowledging the primary strength of our work. We understand the reviewer's concerns, and we aim to address each of them carefully.
>
> We denote the comments as W:weakness, DC:detailed comment and Q:question.
>
>
> ## W1, DC3: Findings under larger ViT
>
> We thank the reviewer for this comment. We run further experiments under two settings; one with a larger backbone but ImageNet checkpoint (iBOT ViT-B/16), and another with a larger backbone but a histopathology pre-trained checkpoint (Phikon ViT-B/16).
> - We show these results in **Appendix B.3 in the revised manuscript**, and a copy of the table is shown below.
> 	**Table: Camelyon16 results under different encoder settings**
> 	“Patch” denotes the time taken (under inference mode) to pre-compute patches of the training samples of the downstream dataset.
>
> 	| Method         | iBOT-B AUC (%) | Phikon AUC (%) | Patch (min) | Training (min) | Total (min) |
> 	|---------------|----------------|---------------|-------------|----------------|-------------|
> 	| ACMIL (Frozen)| 88.08 ± 0.7    | 99.4 ± 0.3    | 48.5        | 6.7            | 55.2        |
> 	| eWSI_64       | 91.61 ± 2.2    | 99.0 ± 0.3    | -           | 45.8           | 45.8        |
>
> - We observe a slight improvement with iBOT-ViT-B and similar performance with the Phikon encoder. This suggests that when an encoder is already strongly pre-trained (e.g., on 40M TCGA patches), additional task-specific adaptation may provide limited gains. A similar trend was observed with Hist-SSL (ViT-S pre-trained on 4M TCGA patches). However, this does not rule out the usefulness of such methods for domain adaptation, especially when aiming for more compute-efficient ways to adapt models to histopathology without large-scale pre-training.
> - When using iBOT-B, the performance gain is modest and lower than that of iBOT-S (iNet-SSL in our work). One possible explanation is that the parameter settings, such as the LoRA rank $r=4$, may not be optimal for larger encoders. As shown below in W2, performance is sensitive to the choice of rank.
>
>
> ## DC1: On std values being 0.0
>
> - This was due to single seed experiments on larger sampling rates. We have now performed experiments for 5 random seeds are reported them in **Table 1** in **page  8 of the revised manuscript**.
> - Some of the std values are still 0.0. This is because the std cannot be found in reported results in the original studies, and we have clarified this in the caption of Table 1. However, in all of our own implementations, we have now performed experiments for 5 random seeds.

---

> > ### Author Response · Authors · 2026-01-24
> > **W2: Lora analysis and W3: Linmax aggregator**
> >
> > ##  W2: LoRA analysis
> > We thank the reviewer for this comment. We address this comment in two parts, first discussing the efficiency-trade off of the LoRA rank $r$, and then the ablation of $r$.
> > - We would first like to clarify that the computational cost reduction by LoRA ($r=4$) instead of full fine-tuning (which is the equivalent of $r=384$ on ViT-S), is minimal in comparison to the computational cost reduction by stochastic sampling of patches. During our experiments, we found the overall training time to be the same with full fine-tuning, and a negligible increase in GPU memory vs. increase in LoRA rank.
> > - In theory, the compute reduction from LoRA compared to full fine-tuning is minimal relative to the savings from sparse sampling, since gradients still backpropagate through the full ViT encoder. The computational time cost of sampling increases linearly with sample rate $M$, as shown in **Table 5 of the revised manuscript**.
> > - We performed ablations on the LoRA rank $r$, and found that increasing $r$ leads to slightly degraded performance, and $r=4$ to be an optimal rank assignment, as shown in **Figure 3 of the revised manuscript**. We discuss the effect of rank $r$ in **page 10 of the revised manuscript**, in addition to the existing analysis on the choice layers to adapt.  In summary, we find that the LoRA component in eWSI mainly acts as an implicit regularizer via its low-rank constraint, rather than a mechanism for reducing computational cost.
> >
> >
> > ## W3: Linmax aggregator
> >
> > We thank the reviewer for providing us the opportunity to explain this further. The LinMax aggregator is motivated specifically by the need to perform end-to-end fine-tuning under sparse sampling regimes.
> >
> > - Recent MIL approaches rely on a **process**$\rightarrow$**pool**$\rightarrow$**process** paradigm. For example, ABMIL and ACMIL first perform a linear projection, then a weighted pooling, and then process the features for classification. However, we find that these methods are not robust to sparse sampling, as shown in  **Figure 2 in the original and revised manuscript**. Here, we experimentally find that max-pooling (FC-Max) is robust to sparse sampling rates, and investigate how we can further improve the richness of features prior to pooling.
> > - We threfore propose LinMax, which **stack** the layers with **gated    activations**, and show that LinMax performs exceptionally well under  sparse sampling regimes, as shown in **Figure 2 in the original and revised manuscript**.
> > - An alternative to LinMax would be stacking the layers with activations as above, and performing weighted-pooling    (i.e.: ACMIL) instead of max-pooling. We perform this experiment with eWSI on Camelyon16, $M=64$ patches for seed=0, and observe AUC=75.5% for iNet-Sup, AUC=78.2% for iNet-SSL, and AUC=97.6% for Hist-SSL, which are worse than max-pooling.
> > - Further, in **Appendix B.4**, we show that simply stacking layers without gating results in sub-par performance. This indicates the need for feature selection via gating, resulting in richer features passed to the pooling function (in this case, max-pooling).
> > - Overall, Figure 2, Appendix B.4 and the above experiment comparing max-pooling vs. weighted pooling given the stacked-gated layers, indicate that **max-pooling is important to maintain robustness** to sparse sampling, while **stacked gated layers contribute towards better feature selection prior to pooling**.

---

> > > ### Author Response · Authors · 2026-01-24
> > > **DC2, Q1: Computational cost , DC4, Q2: Camelyon16 failure cases and sparse sampling**
> > >
> > > ## DC2, Q1: Computational cost
> > >
> > > We thank the reviewer for this comment, and we address the concern of computational cost in **pages 10, 11 of the revised manuscript**. Specifically, we compare eWSI vs. frozen encoder fine-tuning, and compare eWSI vs. existing end-to-end methods.
> > >
> > > **eWSI vs. frozen-encoder fine-tuning:**
> > >
> > > - There are two compute costs associated with frozen-encoder fine-tuning. The first is offline patch extraction on the downstream dataset training samples. For Camelyon16, this was approximately 26 minutes with a Pytorch DataLoader and 10 parallel processes to load data, run on a single RTX 3090 GPU under model inference mode. Once the patches are pre-computed, the cost is negligible, as the total training time is 4 minutes for patch representations of $dim=384$.  This results in a total wall time of approximately 30 minutes.
> > > - With eWSI, patches representations are computed on the fly-during training in order to train the encoder. With $M=64$ and ViT-S, this results in 15 seconds per epoch, which equates to approximately 25 minutes total training time.
> > > - Even if the feature pre-computation time is not considered, 25 minutes on a small-scale GPU remains a practical alternative to frozen-encoder fine-tuning, especially given the performance increase and the ability to quickly move from an ImageNet checkpoint to a domain-and-task specific encoder state.
> > >
> > > **eWSI vs. end-to-end strategies:**
> > >
> > > - When comparing with existing end-to-end methods, StreamingCLAM reports a runtime of 165 seconds per forward pass, making a direct comparison unnecessary. Therefore, we do not include StreamingCLAM in our efficiency comparisons.
> > > - The closest method in terms of efficient end-to-end training is C2C, which only passes a fixed number of pre-defined patches during training (not random). However, C2C still requires extracting features from the full training set at the start of each epoch, which introduces the additional overhead mentioned above.
> > > -	Inference cost remains the same across methods, since all approaches ultimately need to extract features from all patches of the test WSIs for evaluation, regardless of whether this extraction is performed before or after training.
> > >
> > > -	We tabulate the computational cost in **Table 5 of the revised manuscript**, and its copy is shown below.
> > >
> > > 	 Compute cost comparison.
> > > 	 p × 10⁹: number of parameters; M: number of patches; V_gpu: peak GPU memory (GB); t_ep: time per epoch in mm:ss; T: total time for 100 epochs in hours (including feature extraction).
> > > 	 | Method   | p × 10⁹ | M    | V_gpu | t_ep   | T(h)
> > > 	 |----------|---------|------|-------|--------|--------|
> > > 	 | Frozen   | 0.05    | 8192 | 0.03  | 00:02  | 0.48  |
> > > 	 | eWSI    | 0.27    | 64   | 1.10  | 00:15  | 0.42  |
> > > 	 | eWSI    | 0.27    | 384  | 6.58  | 01:12  | 2.00  |
> > > 	 | eWSI    | 0.27    | 512  | 8.82  | 01:31  | 2.53  |
> > > 	 | C2C     | 21.67   | 64   | 1.10  | 25:04  | 40.8  |
> > >
> > > ## DC4, Q2: Camelyon16 failure cases and sparse sampling
> > >
> > > We thank the reviewer for this comment. To address this, we provide further insight into failure cases, and discuss the relationship between prediction confidence and tumor region. We discuss this with the required visualizations in **Appendix B.1 in the revised manuscript.**
> > >
> > > - We first investigate how sparse sampling during training affects small ROIs in test WSIs. We perform patch-wise classification prior to max-pooling, to obtain patch-wise tumor predictions. The threshold to classify a slide as positive or negative is computed via Youden's thresholding. Using the thresholding, we observed 4 false negative slides with eWSI iNet-SSL encoder, shown in **Figure 5 in page 16 (Appendix B.1) of the revised manuscript.**
> > > - We observe that even in the samples wrongly classified as negatives, there exists a high probability of tumor prediction at the true location of the tumor.
> > >
> > > - We further investigated the relationship between tumor burden and prediction confidence. As the raw prediction probabilities are systematically biased, we focus on the association between the two entities. Specifically, we measure the association between tumor percentage and slide-level prediction probability using Spearman’s rank correlation. We observe a strong positive correlation ($\rho = 0.78$), indicating a significant monotonic relationship between tumor burden and prediction probability.

---

### Author Rebuttal · Authors · 2026-01-24

**Rebuttal:**

Dear reviewers,

We thank you for your thoughtful comments and for acknowledging the strengths of our work.

We address each concern in detail in the official responses. In addition, the revised manuscript includes several new analyses and clarifications, including expanded implementation details, ablations on LoRA rank, computational cost, and an analysis of the effect of sampling rate on downstream performance.

Please note that in the revised manuscript, the cyan and light yellow highlighting correspond to corrections and new additions, respectively.

We are happy to clarify any further concerns.

Thank you,
Authors

**Supporting Material:**

/attachment/4c8d8d7831855c77c6aac5ae853f745d01b851f9.pdf

---

> ### Author Response · Authors · 2026-02-01
>
> Dear Reviewers,
>
> With approximately one day remaining in the discussion period, we would like to kindly note that we are happy to clarify any remaining questions regarding our rebuttal.
>
> We believe the rebuttal has addressed the main concerns raised, in particular:
>
> - We demonstrate that the overall compute cost is practical and substantially more efficient than existing end-to-end methods.
> - We show that the limitation of sparse sampling during training in identifying small tumor regions does not compromise the model’s discriminative capacity.
> - We experimentally validate the effectiveness of LinMax (gated stacking combined with max-pooling) compared to existing pooling strategies.
> - We add full implementation details and have released the codebase to support reproducibility.
>
> We hope that the revised manuscript and rebuttal have fully addressed the reviewers’ concerns, and that the strengthened quality of the work is reflected in the final evaluation.
>
> Thank you,
> Authors

---

### Meta-Review · Area_Chair_6X61 · 2026-02-10

**Recommendation:** Accept (Poster)
**Confidence:** 4

**Metareview:**

This paper introduces eWSI, combining PEFT (LoRA) with a LinMax MIL aggregator to enable single‑GPU end‑to‑end WSI training, and shows competitive or superior performance to methods using costly in‑domain pretraining across Camelyon16, TCGA, and BRACS. Reviewers appreciate the strong empirical results, practical impact, and added analyses (compute cost, sparse sampling bias, larger encoders, LoRA rank, LinMax justification), and their ratings converge to weak accept after rebuttal. While the approach is largely ViT‑centric and LinMax is conceptually close to prior max‑pooling MIL, the work still offers a clear, reproducible and useful recipe for efficient task‑specific WSI adaptation.

---

### Decision · Program_Chairs · 2026-02-13

Accept (Poster)